# LMRL-Gym: Benchmarks for Multi-Turn Reinforcement Learning with Language Models

## Abstract

Large language models (LLMs) provide excellent text-generation capabilities, but standard prompting and generation methods generally do not lead to intentional or goal-directed agents and might necessitate considerable prompt tuning. Even the best current LLMs rarely ask clarifying questions, engage in explicit information gathering, or take actions that lead to better decisions after multiple turns. Reinforcement learning has the potential to leverage the powerful modeling capabilities of LLMs, as well as their internal representation of textual interactions, to create capable goal-directed language agents. This can enable intentional and temporally extended interactions, such as with humans, the emergence of complex skills such as persuasion, and long-horizon strategic behavior, such as in the context of games. Enabling this requires the community to develop reliable reinforcement learning algorithms for training LLMs. Developing such algorithms requires tasks that can gauge progress on algorithm design, provide accessible and reproducible evaluations for multi-turn interactions, and cover a range of task properties and challenges in improving reinforcement learning algorithms. Our paper introduces the LMRL-Gym benchmark for evaluating multi-turn RL for LLMs, together with an open-source research framework for getting started on multi-turn RL with offline value-based and online policy-based RL methods. Our benchmark consists of 3 Interactive Dialogue tasks and 5 RL Capability tests for a total of 8 tasks, which require multiple rounds of language interaction and cover tasks in open-ended dialogue and text games.

## 1 Introduction

Large language models (LLMs) have demonstrated remarkable abilities when naturally conversing with humans (OpenAI, 2023; 2022; Touvron et al., 2023; Google, 2023), answering questions and responding to requests (Shuster et al., 2022b;a; Qin et al., 2023), and even performing coding tasks (Chen et al., 2021b; Wang et al., 2023b). Many of these capabilities are enabled by learning to emulate humans from large datasets of text from the web (Völske et al., 2017; Shuster et al., 2022a; Yao et al., 2023), learning from examples "in context" (Brown et al., 2020), as well as learning from other sources of supervision such as instruction datasets (Mishra et al., 2022; Wei et al., 2022; Wang et al., 2022b) and preference fine-tuning with RLHF (Ziegler et al., 2020; Ouyang et al., 2022). However, directly applying LLMs in settings that require planning or multi-turn interactions presents new challenges. LLMs are not explicitly goal-directed, as they are not optimized to directly solve particular tasks, but rather to produce text that resembles the distribution of human-provided examples or accords with human preferences (Ziegler et al., 2020; Stiennon et al., 2020; Wu et al., 2021; Bai et al., 2022a). This challenge is apparent in solving temporally extended tasks, such as multi-turn dialogue (Irvine et al., 2023; , FAIR), complex tool use (Wang et al., 2022a), multi-step games (Hendrycks et al., 2021b), and other interactive applications. In principle, LLMs should contain the knowledge necessary to succeed in such settings: if the multi-turn interactions center around problem domains that are well represented in the model's training data (such as dialogue), well-trained LLMs should already serve as powerful predictive models in such settings. However, leveraging this predictive knowledge to derive effective actions and strategies requires not just emulating humans, but also planning and optimization.

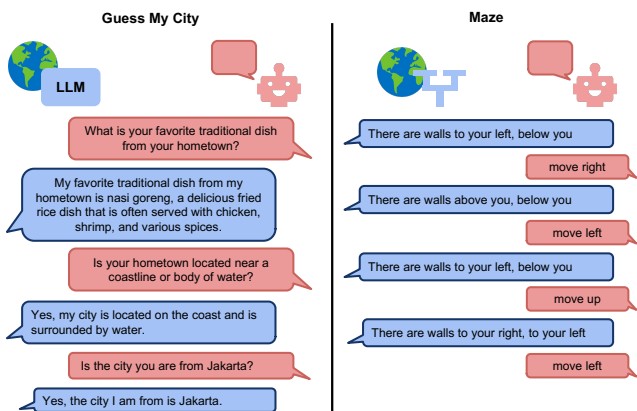

Figure 1: **Overview of LMRL-Gym:** We show sample trajectories from two tasks in our benchmark. In the Guess My City task, the agent learns to ask questions to guess the city the oracle is from, while in the Maze task, the agent learns to make the correct moves based on cues from the oracle.

Multi-turn reinforcement learning (RL) (Sutton & Barto, 2018) in principle offers a path to enable LLMs to do just that. RL could enable goal-directed reasoning and planning in interactive multi-turn settings, including complex dialogue, games, and tool use. We hypothesize that RL could serve as a powerful tool for LLM training, not only for training models to accord with human preferences, but more generally to accomplish tasks in an intentional and goal-directed manner. Text generation can be viewed as a sequential decision-making process, treating a sequence of tokens as a trajectory. Many tasks, such as successfully answering questions or eliciting a desired reaction from a user, can then be framed as optimizing some reward function over these trajectories. However, despite extensive interest in RL for LLMs in recent years, much (though not all) of the recent research in this area has focused on "single-step" RL problems, where a single response is optimized for some quality metric, typically derived from human preference signals (Stiennon et al., 2020; Ziegler et al., 2020; Ouyang et al., 2022; Bai et al., 2022a; Anthropic, 2023; Ramamurthy et al., 2023; Christiano et al., 2023; Casper et al., 2023).

While some works have sought to apply RL for multi-turn tasks (Singh et al., 1999; Li et al., 2016; Shah et al., 2016; Kwan et al., 2022), particularly for goal-directed dialogue (Lewis et al., 2017; Verma et al., 2022), there has been comparatively little research on improving the underlying RL algorithms and very little head-to-head comparison on same sets of tasks. This is perhaps unsurprising: it is easier to evaluate improvements to algorithms for single-turn text generation as compared to multi-turn generation. Multi-turn dialogue requires an interactive evaluation procedure rather than just a static dataset. There is no established protocol for such evaluations, and the "gold standard" constitutes costly and time-consuming studies with human participants.

In this work, we aim to address this challenge and make it possible for RL algorithm researchers to iterate on developing better RL methods for multi-turn language-based interaction tasks, such as dialogue and games. We posit that benchmarking RL algorithms for LLMs presents a very different set of challenges and merits a different set of solutions compared to other benchmarks in NLP. While most NLP benchmarks are based on standard supervised machine learning paradigms, with a training set and a test set (Marcus et al., 1993; Tjong Kim Sang & De Meulder, 2003; Socher et al., 2013; Rajpurkar et al., 2016; Wang et al., 2019; Williams et al., 2018), RL benchmarks require simulators that the trained agents can interact with to measure their performance. In this paper, we use an LLM to simulate a conversation partner in dialogue tasks. While the behavior of the LLM may deviate from human behavior, we verify in a human study in Appendix A that our LLM simulators produce natural text reflecting human norms of conversation. However, our goal is *not* to utilize this approach to benchmark whether LLMs are *good at talking to humans*, but rather as a way to test RL algorithms with datasets that are sufficiently difficult and complex to gauge how effective they might be if they were then trained on data from real humans. Specifically, our benchmark aims to rigorously stress-test the ability of RL algorithms to enable complex goal-directed behaviors in LLMs. To this end, LMRL-Gym also includes a set of text-based strategy games, in addition to the

dialogue tasks, that are aimed at providing a more controlled and focused diagnostic assessment of specific RL capabilities.

Our proposed benchmark, LMRL-Gym, consists of 8 tasks. Three tasks are Interactive Dialogue tasks designed to simulate real-world interactions with humans, requiring information gathering (20 Questions, Guess My City) and negotiation (Car Dealer). Five tasks are RL Capability Tests, which are text games designed to isolate specific capabilities of RL training. Each task comes with an offline dataset that can be used for offline RL training, and a "simulator" that can be used to evaluate the performance of the agents in multi-turn interactive tasks. We provide a research framework and toolkit for researchers and practitioners to get started with multi-turn RL for LLMs. This framework includes implementations of PPO (Schulman et al., 2017), ILQL (Snell et al., 2022a), and several baseline methods, implemented in an extensible way designed for future development of tasks, experimentation, and algorithm design.

## 2 RELATED WORKS

**Datasets, benchmarks, and libraries.** Benchmarks and datasets have been an important factor for driving progress in NLP in domains that include machine translation (Tiedemann, 2012; Bojar et al., 2016), natural language understanding (Rajpurkar et al., 2016; Wang et al., 2019; Hendrycks et al., 2020; 2021a; Ramamurthy et al., 2023), and solving math problems (Cobbe et al., 2021). However, these tasks generally do not involve multi-turn interaction and do not come with rewards, making them hard to adapt to RL research. For example, the standard for evaluating dialogue agents has been to run a human subjects study, but this is time-consuming and costly. Some works have proposed text games for evaluating language-based agents (Chevalier-Boisvert et al., 2018; Hausknecht et al., 2019; Yuan et al., 2019; Fan et al., 2020; Hausknecht et al., 2020; Guo et al., 2020; Ammanabrolu et al., 2020; Yao et al., 2020; Hendrycks et al., 2021b; Singh et al., 2021; Wang et al., 2022a; Yao et al., 2022; Jansen & Côté, 2022; Yao et al., 2023; Zhang et al., 2023; Gontier et al., 2023) and interactive dialogue (De Bruyn et al., 2022b;a). Our aim is to cover a variety of problem settings that reflect challenges in open-vocabulary interaction in addition to text games, that also specifically evaluate offline RL capabilities, which is not done by prior works. Motivated by successes in using LLMs to generate synthetic data (Hausknecht et al., 2019; Park et al., 2023; Bai et al., 2022b), our proposed tasks are based on synthetic data. While such data may differ from natural text, the scope of our benchmark is specific to evaluating RL algorithms, not the ability to interact with humans.

**RL for language models.** RL for language models has seen success in aligning LLMs with human preferences (RLHF) (Ziegler et al., 2020; Stiennon et al., 2020; Bai et al., 2022a;b; Ouyang et al., 2022; Christiano et al., 2023), optimizing non-differentiable objectives for machine translation (Wu et al., 2016; Nguyen et al., 2017; Kiegeland & Kreutzer, 2021), generation (Tambwekar et al., 2019; Pang & He, 2021; Pyatkin et al., 2022), dialogue (Cuayáhuitl et al., 2015; Georgila & Traum, 2011; Li et al., 2016), question answering (Pyatkin et al., 2022), and summarization (Paulus et al., 2017; Böhm et al., 2019; Wu & Hu, 2018). These include RL methods that learn by directly interacting with the environment (online RL) (Carta et al., 2023) and RL methods that only use a static dataset (offline RL) (Jaques et al., 2020; Snell et al., 2022a; Jang et al., 2022; Verma et al., 2022; , FAIR). However, many of these works operate in the singe-step bandit setting, and do not consider multi-turn goal-directed tasks. Our benchmark, on the other hand, focuses on tasks involving multiple turns of interaction with clearly defined goal-based reward functions.

**Capabilities of LLMs.** There has been a surge in the capabilities of LLMs for generation (Ghazvininejad et al., 2017; Radford et al., 2019), dialogue (Lewis et al., 2017; Jaques et al., 2017; Shuster et al., 2022b; Snell et al., 2022b), question answering (Pyatkin et al., 2022), summarization (Paulus et al., 2017; Böhm et al., 2019; Wu & Hu, 2018), text-based games (Narasimhan et al., 2015; Hausknecht et al., 2019), translation (Gu et al., 2017), and more. However, these are often supervised learning tasks that do not test the LLMs' abilities to achieve a specific long-term objective. Research on dialogue generation (Jaques et al., 2017; He et al., 2018; Shuster et al., 2022b;a) has often focused on generating feasible-looking agent dialogue without explicit consideration for some multi-turn objective. Our benchmarks allow for the development of algorithms that enable LLMs to *interact* with an environment to achieve long-term objectives, by providing tasks with online simulators and offline datasets.

## 3    MULTI-TURN GENERATION WITH RL AND LANGUAGE MODELS

This section introduces the conceptual foundations of using reinforcement learning for multi-turn generation with language models. We introduce a definition of the Markov decision process for language and a framework for the methods we focus on in this paper.

**Definitions.**    We formalize language generation tasks as a partially observable Markov decision process. We define the state to be the history of tokens and an action as the next token generated by the model. An observation is a single token $o_i$ in the history. The probability of generating the next token is dependent on all of the previous observation tokens $o_i$. Therefore the Markovian state $s$ is formed by the concatenation of all the previous tokens $[o_0, \ldots, o_i]$. A policy $\pi$ defines the agent's behavior by taking in the current state $s$ and outputting a new action token $a$ to get $s_{i+1}$. The environment assigns a reward $r(s, a)$ based on the entire sequence of tokens so far. The tokens in the state are either generated by the policy $\pi$ or the environment. For example, in the Car Dealer task, the policy generates the tokens for the Seller's utterance and the environment generates the tokens for the Buyer. The full history of their conversation would form the state. A complete sequence of tokens is referred to as a trajectory $\tau = o_0, \ldots, o_T$. The goal of RL is to produce a policy $\pi^*$ that maximizes the expected discounted sum of rewards over trajectories $(\tau)$ under the policy $\pi^* = \arg\max_\pi \mathbb{E}_{\tau \sim \pi} \left[ \sum_{t=0}^{T-1} \gamma^t r_t(s_t, a_t) \right]$, where $\tau$ represents the trajectory.

**RL Algorithms.**    Several possible RL algorithms could be used to train language models for multi-turn tasks (Jaques et al., 2020; Verma et al., 2022; Snell et al., 2022a; Schulman et al., 2017; Stiennon et al., 2022; Bai et al., 2022a; Casper et al., 2023). Policy gradient methods, such as PPO (Schulman et al., 2017), directly compute the gradient of the RL objective with respect to the model parameters. Value-based methods estimate a state-action ($Q$) and/or state-value ($V$) function. The state-action or state-value function forms a policy by either 1) acting greedily with respect to the Q-function or 2) perturbing the base model's logits with the learned action-value functions (Snell et al., 2022a). RL methods for training LLMs can be *online* or *offline*. Online methods repeatedly interact with the environment, collecting additional data during training. Offline RL instead learns to extract the best behaviors from an existing, potentially suboptimal dataset. Due to the large amount of existing text interactions on the internet, offline RL is an ideal setting for training language models. Therefore, our work primarily focuses on benchmarking offline RL algorithms. However, our tasks also fully support online RL and we include an online PPO baseline in our evaluation.

## 4    THE LMRL-GYM: SYNTHETIC BENCHMARKS FOR RL WITH LANGUAGE

Our benchmark consists of 8 tasks grouped into two categories: RL Capability tasks and Interactive Dialogue tasks. The RL Capability tasks focus on desirable capabilities for RL algorithms for LLMs such as strategic decision-making, credit assignment, trajectory stitching, partial observability, and use of complex language. For the interactive dialogue tasks, we model them after real-world interactions with humans, such as persuading someone to buy a car or playing a guessing game.

Below, we define the Interactive Dialogue tasks, describe the specific capabilities of RL algorithms for LLMs that our benchmark aims to evaluate through RL Capability tasks, and summarize the data generation and simulation process. We have provided example trials for each task are shown in Figure 4, and a concise summary of the dataset and task statistics in Table 1. The number of trajectories and the average length of the trajectories varies based on the complexity of the tasks.

### 4.1    INTERACTIVE DIALOGUE TASKS

The Interactive Dialogue Tasks aim to simulate real-world goal-oriented dialogues. We focus on tasks where the agent must make inferences about persuasive strategies and actively gather information by asking questions. Instead of generating these interactions with humans, we generate such interactions through simulating LLMs inspired by successes in using LLMs to generate synthetic data. While the LLM might not be as realistic as a real human, we have found that human raters evaluated the LLM-generated text as quite realistic in most cases, as discussed in our user study in Appendix A. You can find examples from the trained models in Appendix I.

**20Qs (Twenty Questions).** This task tests whether an agent can gather information about an unknown subject through twenty yes or no questions. The agent must use semantic knowledge of the object to infer the correct answer.

**Guess (Guess My City).** The Guess My City task performs more complex forms of information gathering, involving open-ended questions about a city. This task evaluates semantic knowledge of a specific city and the agent's ability to parse information from a free-form answer.

**Car Dealer.** The Car Dealer task tests the ability of RL algorithms to learn successful car sale strategies. This involves decision-making and credit assignment as different persuasion strategies must be adopted for different kinds of buyers.

## 4.2 RL CAPABILITY TASKS

A central objective of our benchmark is to evaluate the core capabilities that RL enables in large language models. The RL Capability tasks are text-based games designed to isolate specific RL capabilities and are language analogs of tasks where RL is known to succeed. These tasks include Chess, Endgames, Wordle, Maze, and Text-Nav. Below we explain the tasks and the motivation for including them as tests for RL capabilities. Further details on task design for RL Capability tasks can be found in Appendix B.

|  | Strategic Decision Making | Complex Language | Credit Assignment | Partial Observability | Trajectory Stitching |
|---|---|---|---|---|---|
| Maze FO | ✗ | ✗ | ✓ | ✗ | ✓ |
| Maze PO | ✗ | ✗ | ✓ | ✓ | ✓ |
| Text-Nav FO | ✗ | ✓ | ✓ | ✗ | ✓ |
| Text-Nav PO | ✗ | ✓ | ✓ | ✓ | ✓ |
| Wordle | ✓ | ✗ | ✗ | ✓ | ✓ |
| Chess | ✓ | ✗ | ✓ | ✗ | ✓ |
| Endgames | ✓ | ✗ | ✓ | ✗ | ✓ |

Figure 2: We have designed our RL Capability tasks as text games that include Chess, Endgames, Wordle, Maze, and Text-Nav. These tasks isolate some subset of the RL Capabilities outlined in Appendix B.1.

**Desirable RL capabilities.** RL shines in goal-directed tasks that require multi-step planning and *strategic decision-making*. Strategic decision-making can range from asking follow-up questions (e.g. 20 Questions), to complex strategy in chess. In RL, it is necessary that algorithms can properly perform *credit assignment* as rewards are often delayed relative to the action pivotal to the outcome. A challenge with optimizing POMDPs is *partial observability*, where the agent must make deductions based on incomplete information. In the offline RL setting, the ability of algorithms to perform *trajectory stitching* is often desirable for learning optimal policies from suboptimal trajectories. Lastly, when working with language models, it's important that algorithms remain effective in the face of *complex language* with open-ended generation. We design our RL-capability tests with the goal of stress-testing each of these capabilities, as shown in Figure 2.

**Maze and Text-Nav.** We consider a Maze task as well as the Text-Nav featuring more complex language. Though Text-Nav involves stochastic language, the maze task has longer dataset trajectories and a more complicated layout. To test partial observability, we include both a partially observed and fully observed version of each task. In the partially observed version, we remove information from the maze description such that the agent must infer its position from its move history. To emphasize the comparison to a non-text-based version, we evaluate the Maze task in a symbolic or grid-based environment seen in Appendix H.

**Strategy games.** We include three strategy games; Wordle, Chess, and Endgames. Wordle tests partial observability over the space of possible words while Chess and Endgames test the ability of the agent to form longer-term plans. Endgames provide a simpler and more goal-directed variation of the Chess task. By focusing on the endgame, we encourage algorithms to learn strategy rather than memorizing the opening moves of a chess game. A classic theoretical endgame position consists of a position where the only pieces on the board are the two kings and the queen. All RL Capability tasks evaluate *trajectory stitching* capability through the inclusion of suboptimal trajectories. Further details about our dataset generation strategies can be found in Appendix D. The Chess, Endgames, Maze and Text-Nav tasks test *credit assignment*, because the RL algorithm must learn to assign credit to good actions rather than a lucky starting position in the maze task, or a weak opponent moves in the Chess or Endgames task.

## 4.3 AN OVERVIEW OF DATA COLLECTION FOR LMRL-GYM

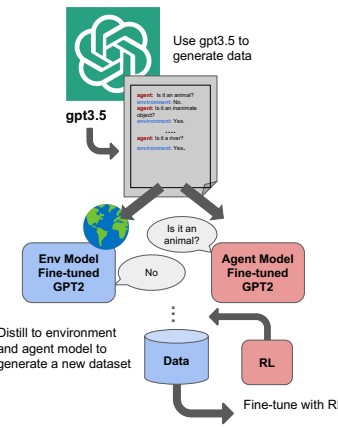

Figure 3: To generate data for conversational tasks, we use LLMs as "simulators" for the task. Our simulators can be used to generate offline data, to provide a "simulation environment" for evaluation, to perform online training, and to compute rewards.

To make tasks in LMRL-Gym practical for benchmarking RL methods, we must balance accessibility and realism. As RL algorithms need to be evaluated by running a learned policy, real-world tasks are comparatively inaccessible for rapid iteration (e.g., if they require talking to real humans). We therefore use simulators for our tasks, derived either from text-based games, or conversational agents powered by language models. Although this fully synthetic setup sacrifices the realistic nature of tasks, we believe significant gain in accessibility is worthwhile and will enable rapid RL algorithm progress.

**RL Capability tests.** For each task, we use a simulator such as a chess engine or maze solver to generate near-optimal data and then we dilute the policy with suboptimal data by taking suboptimal actions or using inferior policies. We also convert our task from a symbolic version to a text-based version in a programmatic way as discussed in Appendix B.

**Interactive Dialogue tasks.** For conversational tasks, we leverage existing LLMs to generate our data, either with two instances of LLMs "talking" to one another or all at once through few-shot prompting as shown in Figure 3. To train these LLMs, we use OpenAI's GPT-3.5 to generate an initial dataset by asking reasonable questions and answers out-of-the-box, collecting a dataset of differing sizes depending on the task. In the case of 20Qs and Guess My City, we collected 1K conversations by querying GPT-3.5 (text-davinci-003) to generate both sides of the conversation based on specific prompts (which can be found in Appendix D.6. To generate the dataset for training our algorithms, we fine-tuned a FLAN-T5-XL guesser model and a FLAN-T5-XL oracle model on their respective sides of the conversation. Using these distilled models, we generated a new dataset of 100K conversations by having the two models talk to each other. We conducted a similar process for the Car Dealer task but with a larger model for fine-tuning (GPT2-XL). When generating our datasets, we also spent considerable effort to ensure diversity in the responses to ensure the collection of high-quality data. For the Car Dealer task as an example, this included providing different desired brands, features, classifications (i.e. car or truck), and budgets in our prompting to generate the datasets. Further details on our data generation process for the three Interactive Dialogue tasks can be found in Appendix D.

| Task | 20Qs | Guess | Car | Maze | Text-Nav | Wordle | Chess | Endgames |
|---|---|---|---|---|---|---|---|---|
| Size | 100k | 100k | 19k | 1.24k | 2.5k | 1m | 625k | 97.756k |
| avg length | 14.9 | 18.8 | 16.5 | 19.7 | 12.2 | 4.82 | 46.7 | 11.9 |
| std length | 4.38 | 4.57 | 3.61 | 24.5 | 8.77 | 1.27 | 18.16 | 12.0 |
| success rate | 0.31 | 0.53 | 0.53 | 0.11 | 0.26 | 0.70 | 0.60 | 0.59 |
| avg return | -17.3 | -18.8 | 0.562 | -19.7 | 0.258 | -4.12 | 0.210 | 0.586 |
| std return | 2.56 | 4.12 | 0.422 | 24.5 | 0.424 | 1.59 | 0.970 | 0.492 |

Table 1: Statistics for all tasks in LMRL-Gym. Size represents the number of trajectories, the average length is the average length of trajectories in the dataset where the unit is a response from the agent. The success rate is the proportion of trajectories that reach the objective. Finally, the reward functions for each task are defined in Appendix D.

## 5 LMRL-GYM RESEARCH FRAMEWORK FOR ALGORITHM DEVELOPMENT

We evaluate the LMRL-Gym tasks on both online and offline RL algorithms, including variations of behavior cloning, value-based RL methods, and online PPO. We have selected these algorithms have they are currently the state-of-the-art methods RL methods for LLMs Chen et al. (2021a); Snell et al. (2022a); Ouyang et al. (2022). With these experiments, we expect to observe (1) a significant spread in performance between the different algorithms, highlighting differences between RL algorithms; (2)

room to improve beyond, such that our benchmark can enable future algorithmic development. Our project page (REDACTED) contains links to our open-sourced datasets (REDACTED) and research framework (REDACTED).

**BC, Filtered BC, Online Filtered BC.**   In line with standard RL nomenclature, we denote supervised fine-tuning as behavioral cloning (BC). This baseline tests whether LMs can effectively represent the behaviors in the datasets. Filtered BC is identical, except only the most successful examples in the offline dataset are used for fine-tuning, a technique which is also used in Snell et al. (2022a). Online filtered BC collects data online using the current policy and selects the most successful trajectories for finetuning. See Appendix E for our data filtering criteria for each task.

**Offline Value-based RL: MC Returns and ILQL.**   Monte-Carlo returns (Kakutani, 1945) and Implicit Language Q-Learning (Snell et al., 2022a) train a value $V$ and $Q$ function. In MC Returns, we train the $Q$ function with an MSE to predict the reward-to-go. In ILQL we train the two action-value ($Q$) functions using the Bellman backup operator (Kostrikov et al., 2021). For both algorithms, the $Q$ and $V$ functions are then used to perturb the logits of the original BC model (see Equation 5).

**Online RL: PPO.**   PPO (Schulman et al., 2017) is an online RL algorithm widely adopted for training language models with Reinforcement Learning from Human Feedback (Christiano et al., 2023; Stiennon et al., 2022; Bai et al., 2022a; Casper et al., 2023). Unlike previous value-function RL methods, PPO learns a language model policy with no policy extraction step.

**GPT4.**   Few-shot prompting is a common technique for creating interactive language agents Wang et al. (2023a). To compare this to RL fine-tuning we few-shot prompt GPT4 using dataset examples and a detailed explanation of the game for each task. The prompts can be found in our code repository.

**Training and evaluation protocol for algorithms.**   For the BC and filtered BC methods, we initialize our models with the pre-trained GPT2 weights (Radford et al., 2019) and perform standard fine-tuning. We choose GPT2 rather than a larger model due to memory and time constraints, though we admit larger models would lead to a performance boost. For each of the RL methods, we initialize the weights of the base model with the weights from the BC checkpoint and then continue finetuning with the RL objective. When fine-tuning PPO, we limit the number of samples to less than 100k. We report the hyperparameters that we used for each task in Appendix E. We evaluate each policy by measuring the average reward in the simulated environment for each task.

**Evaluation of data generation.**   When using LLMs as a simulator for human actions, it is important to verify that (1) the text produced by the LLM is natural and (2) LLM simulator is not exploitable e.g. policy achieves high reward without actually accomplishing the goal. In addition to validating the data generation process through statistics reported in Table 1, we verified the naturalness of the LLM-produced text in a user study of 40 users. In this study, found no significant difference in the naturalness of conversations generated by ChatGPT3.5 and our trained simulators and agents Appendix A. For example, natural conversations imply that the strategies employed by the Seller to convince the Buyer followed human patterns of conversation and indicate the robustness of the Buyer model to hacking. 20 Questions and Guess My City are particularly hard to hack as they require the agent to successfully guess the word. We verify this through automatic checks as described in our prompting strategy in Appendix D.6.

## 6   BENCHMARKING BASELINE RL METHODS

In Table 2 we present the results for each method on each of our text-game and interactive dialogue tasks. We normalize the scores such that a score of 50 corresponds to the average reward in our offline dataset, 0 corresponds to the lowest possible score, and 100 to the highest score. Across all tasks, we see that our offline RL baseline methods consistently outperform both the dataset and the filtered BC policies, demonstrating the efficacy of offline RL in representing a more optimal policy than the best behaviors in the data. Similarly, we see that online PPO generally improves over the BC policies, highlighting the utility of learning from online environment interaction. However, between RL Capability tasks and Interactive Dialogue tasks, we observe desperate trends in which specific method performs the best. We discuss this in more detail below.

| | alg. | BC | % BC | MC Return | ILQL | Online PPO | Online % BC | GPT-4 |
|---|---|---|---|---|---|---|---|---|
| **Interactive Dialogue** | 20Qs | 57.1 | 77.1 | 87.1 | 82.9 | 72.9 | 55.2 | **95.7** |
| | Guess | 30.0 | 48.0 | 88.0 | 75.0 | 49.9 | 31.6 | **92.3** |
| | Car | 44.5 | 54.8 | **57.2** | 46.3 | 50.5 | 40.4 | 53.5 |
| **RL Capability tasks** | FO Maze | 58.2 | 68.9 | 75.0 | **99.9** | 79.7 | 57.4 | 78.2 |
| | PO Maze | 53.1 | 50.1 | 52.4 | **76.3** | 42.4 | 53.1 | 60.4 |
| | FO Text-Nav | 53.7 | 65.1 | 71.9 | **91.8** | 87.1 | 74.5 | 67.5 |
| | PO Text-Nav | 49.7 | 60.5 | 71.6 | 83.7 | **85.5** | 68.4 | 40.2 |
| | Wordle | 79.9 | 79.1 | 94.9 | **97.7** | 84.2 | 95.2 | 15.4 |
| | Chess | 47.2 | 42.9 | 46.5 | 47.3 | **48.0** | 47.2 | 0 |
| | Endgames | 35.1 | 17.7 | 50.2 | 45.8 | **77.5** | 36.2 | 0 |

Table 2: Normalized reward for all tasks. We present the interactive dialogue tasks on top and the RL capability tasks on the bottom. Value-based methods (MC and ILQL) generally outperform filtered BC, as we might expect in stochastic settings, though the relative performance of ILQL and the simpler MC method is, perhaps surprisingly, reversed on the tasks with more complex language, suggesting that there is room for improvement with such methods. Online RL with PPO often, but not always, improves over offline methods that are not permitted to collect additional online interaction. To make the results more comparable across tasks, we normalize the average return for each policy such that 0 is the minimum possible return, 50 is the dataset average return, and 100 is the maximum return for each task. We also report the raw score results and evaluation details in Appendix F.

**Which algorithm performs best on the RL Capability tasks?** On the RL Capability tasks in Table 2, we see ILQL has the highest performance across all methods for most tasks. ILQL's performance on these tasks is likely due to its unique ability to perform trajectory stitching, enabling it to outperform any individual trajectory in the dataset by learning to compose the best parts of many different trajectories. However, on the PO text-nav, chess, and endgames tasks, we see that PPO outperforms ILQL, suggesting that there is likely still much room for improvement in terms of developing better offline TD-based RL methods for LLMs.

**Which offline RL algorithm performs best for Interactive Dialouge tasks?** In contrast to the text-based games, on our Interactive Dialogue tasks, we see that across all tasks ILQL under-performs the simpler MC returns method. This discrepancy with dialogue, may be because on the more complex text-based tasks it is harder to scale full TD-learning. In fact, we find that on the car-dealer task, even filtered BC outperforms ILQL. Overall, these findings demonstrate that there is much progress to be made in developing better offline RL methods that can effectively optimize LLMs in complex and realistic dialogue settings.

**How does performance of language-based text games compare with their symbolic-based counterparts?** We created a non-text-based version of the Maze task (an RL Capability task) to investigate what difficulties arise from deploying RL algorithms on language-based tasks. We found that simple online and offline Q-learning was able to get an optimal score on the maze. Therefore, the performance symbolic maze is comparable to the fully observed Maze task. However, on the PO Maze task, the language-based methods perform significantly worse. This highlights room for improvement in dealing with partial observability in environments with complex language. Further details for this ablation are found in Appendix H.

**How does prompting GPT-4 compare with RL fine-tuning?** On the RL Capability tasks, we find that our much smaller RL finetuned models significantly outperform GPT4, demonstrating the efficacy of RL for enabling complex goal-directed behaviors in language models. However, on the Interactive Dialogue tasks, GPT-4 outperforms or performs on par with our best RL-trained models. These dialogue tasks are likely to be much more in distribution for GPT4 than our text-game RL capability tasks, and thus GPT4's broad world-knowledge, reasoning, and conversational abilities become synchronized allowing it to compensate for its lack of goal-directed RL fine-tuning in these scenarios. Nonetheless, the mere fact that finetuning small models with RL enables us to close much of the gap to GPT4 on these more realistic tasks underscores the efficacy of RL finetuning. In summary, we can see that RL algorithms consistently outperform baselines like filtered BC on

many of the tasks. However, these results highlight significant areas for growth. For example, the instabilities observed in training PPO require further investigation beyond hyperparameter tuning. Moreover, the performance discrepancy between ILQL and the simpler MC Returns highlights that scaling full TD-learning to Interactive Dialogue settings is another area for improvement.

## 7 DISCUSSION

We propose LMRL-Gym, consisting of 8 tasks including three Interactive Dialogue tasks, and five RL Capability tests. We provide a research toolkit for practitioners to get started with multi-turn RL for LLMs. Our objective is enable the iteration and development of more effective methods for language-based, multi-turn interaction tasks. This includes enabling core capabilities in LLMs through RL to perform complex decision-making, complex conversational interactions, credit assignment, and trajectory stitching. Our evaluation shows promise of RL in several tasks, with further room for improvement with a push for better methods. We acknowledge several limitations when designing tasks in our benchmark, including primarily leveraging smaller GPT-based LLMs to generate datasets and finetune our LLM-based simulators. While we have primarily trained and evaluated models with a maximum 1.5B parameters, we have maintained a lower parameter count to ensure accessibility for researchers with limited computational resources. In addition to releasing our code and datasets, we share all of the hyperparameters we used to train our models in Appendix E and provide more in-depth insight into our results, training procedure, and evaluation in  Appendix F.

We would like to acknowledge that this work is part of a larger effort to improve the performance of LLMs in settings that require planning or multi-turn interactions including multi-turn dialogue, complex tool use, multi-step games, and other interactive applications. Our goal is to propose tasks to evaluate different capabilities expected from an LLM, such as common sense reasoning, credit assignment, reasoning under uncertainty, information-seeking behaviors, and trajectory stitching. We hope this benchmark inspires the creation of more synthetic datasets and simulators for dialogue and is used to design better algorithms to train goal-directed LLM-RL models.

## 8 IMPACT STATEMENT

This work aims to develop a benchmark for the advancement of research in reinforcement learning and LLMs. We generate datasets for tasks in our benchmark with existing LLMs for dialogue tasks and online engines for text games, adhering to best practices in data handling and ensuring there is no personally identifiable or sensitive information present in the generated datasets. We recognize that there may be biases present in the datasets we collect, and have taken steps to ensure a diverse and varied collection of responses from LLMs for our conversational task as detailed in our data generation process in Appendix D. In considering the ethical implications of interactive RL, we acknowledge the dual use implication of this research, particularly centered around developing LLM simulators that could perform persuasion, manipulation, and addictive engagement of users at a large scale. The optimization processes employed by such algorithms, which aim to maximize certain objectives, raise ethical considerations when the optimized outcomes may prioritize system goals over user safety and alignment to human values. We have designed our datasets and reward functions such that prioritize fairness and human-aligned outcomes. By incorporating these considerations when designing our framework, we aim to encourage the development of reinforcement learning models and LLMs that not only excel in performance but also adhere to ethical standards.

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
