# OpenReview forum: "LMRL Gym: Benchmarks for Multi-Turn Reinforcement Learning with Language Models"
_ICLR.cc/2025/Conference — Submitted to ICLR 2025_

### Official Review · Reviewer_KbFz · 2024-11-02

**Soundness:** 2
**Presentation:** 2
**Contribution:** 3
**Rating:** 5
**Confidence:** 4

**Summary:**

The paper highlights that current LLMs are trained to imitate golden responses rather than genuinely learning to reason and solve single-turn tasks. Additionally, there is a lack of benchmarking for multi-turn RL tasks, along with the absence of established evaluation protocols, which can be costly. To address this, the authors synthesize a benchmark that leverages the imitation capabilities of language models in conjunction with simulators, such as chess engines. They propose the LMRL-GYM benchmark, which comprises three interactive dialogue tasks and five RL capability tests, benchmarking existing RL methods, including offline methods like ILQL and online methods like PPO, among others.

**Strengths:**

The paper raises a significant question regarding the benchmarking of different RL algorithms in multi-turn scenarios and introduces the LMRL-GYM benchmark, which consists of several tasks designed for evaluation. It assesses a diverse range of RL algorithms while also providing a comprehensive evaluation framework.

**Weaknesses:**

1. The real-world tasks included in the benchmark are not sufficiently representative, as they only incorporate three tasks that focus on abilities such as persuasion and information gathering.
2. The dataset construction appears somewhat unconvincing. For the interactive dialogue tasks, authors initially use two GPT-3.5 models to generate the dataset and then train two FLAN-T5-XL models to imitate the guesser and oracle roles. Since these are relatively small models, the resulting dialogues may lack diversity and representativeness. The reliability of the benchmarking results for various RL algorithms raises concerns. While the authors conducted a user study to assess the naturalness of the synthesized datasets, I remain skeptical about the benchmark's overall naturalness.
3. The RL ability benchmark, which consists of five tasks, has a limited action space, deviating from real-world scenarios that utilize RL with much larger action spaces, such as step-wise scoring for tasks like math or code generation.
4. The experiments are conducted with small models; is the benchmark applicable to larger models? Since small models can achieve nearly 100 rewards on some tasks (as shown in Table 2), this may impact the significance of the benchmark.
5. In Table 2, the performance of GPT-4 prompting is significantly worse than that of the RL algorithms on the RL capability tasks, despite GPT-4 also being trained using RL methods. Can you comment on this?
6. The right side of Table 1 extends beyond the page margin, and some tables in the appendix exhibit the same issue.

**Questions:**

The questions are outlined in the weaknesses section above.

---

### Official Review · Reviewer_WagK · 2024-11-03

**Soundness:** 2
**Presentation:** 2
**Contribution:** 2
**Rating:** 3
**Confidence:** 4

**Summary:**

1. The paper introduces the LMRL-Gym benchmark for evaluating multi-turn Reinforcement Learning (RL) for Large Language Models (LLMs).
2. The benchmark consists of 3 Interactive Dialogue tasks and 5 RL Capability tests which require multiple rounds of language interaction.
3. A research toolkit for practitioners has been provided to get started with multi-turn RL for LLMs with offline value-based and online policy-based RL methods.

**Strengths:**

A benchmark LMRL-Gym highlighting the importance of multi-turn RL for LLMs has been proposed in the paper. Evaluating multi-turn RL is important for LLMs, and offers future introspection whether RL can generalize for LLMs.

A research toolkit has been proposed for multi-turn RL for LLMs with offline value-based and online policy-based RL. This can be useful to practitioners in the field as an engineering guide.

**Weaknesses:**

Lines numbers have been abbreviated as L# in the points below e.g. L100 means Line 100. Observations have been given quoting paper lines. Some observations are general where no line numbers have been quoted.

1. L074-L075 Multi-turn reinforcement learning (RL) (Sutton&Barto,2018) in principle offers a path to enable LLMs to do just that.

Observation: Sutton & Barto can be cited for general reinforcement learning algorithms and not for Multi-turn reinforcement learning algorithms specifically.
There are other papers which define multi-turn reinforcement learning which have not been cited like "Multi-turn Reinforcement Learning from Preference Human Feedback" https://arxiv.org/abs/2405.14655

2. L086 – L089 While some works have sought to apply RL for multi-turn tasks (Singh et al.,1999;Li et al.,2016; Shah et al.,2016;Kwan et al.,2022), particularly for goal-directed dialogue (Lewis et al.,2017; Verma et al.,2022), there has been comparatively little research on improving the underlying RL algorithms and very little head-to-head comparison on same sets of tasks.

Observation: What does ‘comparatively little research’ and ‘very little head-to-head comparison’ refer to? It should be mentioned why the comparisons in existing multi-turn tasks is not enough

3. Observation: What contribution and value addition does the present work make? It seems that already published papers cover the paper's goals.

4. L100 – L103: In this paper, we use an LLM to simulate a conversation partner in dialogue tasks. While the behaviour of the LLM may deviate from human behavior, we verify in a human study in Appendix A that our LLM simulators produce natural text reflecting human norms of conversation.

Observation: The human study with 40 participants and 18 natural text examples does not statistically justify that the simulation results reflect human norms of conversation. What is the basis of the simulation results reflecting human norms of conversation on a very small sample size of participants and likewise very small number of examples?

5. L105 – …. to test RL algorithms with datasets that are sufficiently difficult and complex to gauge how effective ….

Observation: How do you define datasets that are sufficiently difficult and complex to gauge? Is there any metric or any qualitative decision making? The phrasing "sufficiently difficult and complex" needs to be justified

6. L117 - L118: This framework includes implementations of PPO (Schulman et al.,2017), ILQL(Snell et al.,2022a), and several baseline methods, ….

Observation: What other baseline methods? It should be mentioned in the appendix at least

7. L129 – L130: Some works have proposed text games for evaluating language-based agents and interactive dialogue.

Observation: If other research papers have already proposed text games for evaluating language-based agents and interactive dialogue, please justify why this paper using RL algorithms for such tasks is a novel or a major contribution. Is there any engineering benefit? Please share that as other papers have covered this direction of research.

8.. L205 – L206: We have provided example trials for each task are shown in Figure 4, and a concise summary of the data set and task statistics in Table 1.

Observation: Please correct Grammatical errors like "are shown" should be "as shown". Please note that clicking Figure 4 leads to Figure 1. The source tex file needs to be corrected. Also please mention that Figure 4 is in Appendix B.

10. L321-L322: We have selected these algorithms have they are currently the state-of-the-art methods RL methods for LLMs

Observation: Please revise the sentence construction. The paper needs edits and revisions before publication.

11. L441-L442: Our objective is enable the iteration and development of more effective methods for language based, multi-turn interaction tasks.

Observation: Correction of the phrase ‘is enable’ to ‘is to enable’ should be done

**Questions:**

Lines numbers have been abbreviated as L# in the points below e.g. L028 means Line 28. Questions have been given quoting paper lines. Some observations are general where no line numbers have been quoted.

1. L 028 L031 Our benchmark consists of 3 Interactive Dialogue tasks and 5 RL Capability tests for a total of 8 tasks, which require multiple rounds of language interaction and cover tasks in open-ended dialogue and text games.

Question: How does the Benchmark cover tasks in open-ended dialogue and text games is not described in the paper?

There can be many notions of open-endedness be it in dialogue https://begrijpelijkeformulieren.org/sites/begrijpelijkeformulieren/files/Reja_e.a._Open-ended_vs._Close-ended_Questions_in_Web.pdf or Reinforcement Learning https://proceedings.mlr.press/v119/wang20l.html

Please specify with examples what form of open-endedness has been discussed here.

2. L 047 L049 This challenge is apparent in solving temporally extended tasks, such as multi-turn dialogue (Irvine et al.,2023;,FAIR), complex tool use (Wang et al.,2022a), multi-step games (Hendrycksetal.,2021b), and other interactive applications.

Question: If existing research papers have already covered multi-turn dialogues, complex tool use and multi-step games, then

What is the contribution of this paper by using RL algorithms for multi-turn dialogues, multi-step games? Please share any research or engineering benefits?

3. Question: What are the other interactive applications that has been mentioned in the above L047-L049 lines?

4. Question: How important are RL capability tests for multi-turn RL? How are challenges of RL (generalizability, sample complexity etc) affecting the LLM interaction?

5. Question: Policy Gradients in RL algorithms can be unstable for which different seeds have to be selected. What seeds were selected for the policy-gradient algorithms supported in this work like PPO and other algorithms?

---

> ### Author Response · Authors · 2024-12-04
> **Response to Reviewer WagK**
>
> We thank the reviewer for their feedback and very helpful observations, we appreciate the clarity. We have addressed points raised in your review below:
>
> 1. *"L074 - L075"*: We thank the reviewer for this clarification point and will be sure to make the modification in the paper.
> 2. *"L086 - L089" While some works have sought to apply RL for multi-turn tasks (Singh et al.,1999;Li et al.,2016; Shah et al.,2016;Kwan et al.,2022), particularly for goal-directed dialogue (Lewis et al.,2017; Verma et al.,2022), there has been comparatively little research on improving the underlying RL algorithms and very little head-to-head comparison on same sets of tasks.*
>
> We would like to clarify that the lack of comparisons refers to the fact the RL algorithms have not been compared directly on the same suite of tasks before. So far, these papers have created their own benchmark while proposing an algorithm. For example, the ILQL paper [1] or the NLPO paper [2] both introduce separate tasks, and the paper that introduced the 20 Questions task only benchmarked with the PPO algorithm [3]. We seek to remedy this and provide a suite of tasks and algorithms that we can use to directly compare performance across a variety of RL LLM algorithms. This is a significant contribution because although some of the tasks have been considered before, algorithms to improve performance on these tasks have not been evaluated and directly compared. Moreover, we provide a codebase for training new algorithms in JAX directly on our task.
>
> Works Cited:
> 1. Snell, Charlie, et al. "Offline rl for natural language generation with implicit language q learning." arXiv preprint arXiv:2206.11871 (2022).
> 2. Ramamurthy, Rajkumar, et al. "Is reinforcement learning (not) for natural language processing: Benchmarks, baselines, and building blocks for natural language policy optimization." arXiv preprint arXiv:2210.01241 (2022).
> 3. Zhang, Y., Lu, J., & Jaitly, N. (2024, August). Probing the multi-turn planning capabilities of LLMs via 20 question games. In Proceedings of the 62nd Annual Meeting of the Association for Computational Linguistics (Volume 1: Long Papers) (pp. 1495-1516).

---

> > ### Author Response · Authors · 2024-12-04
> > **Response to Reviewer WagK**
> >
> > 3. "What contribution and value addition does the present work make? It seems that already published papers cover the paper's goals.
> >
> > The goal of our paper is to present a benchmark that applies RL algorithms to multi-turn tasks, specifically to perform goal-directed dialogue. To this end, we provide (1) online simulators and offline datasets for a suite of 7 text-based strategy games and dialogue tasks (2) methodology to create simulators for offline evaluation, online RL training, and computing rewards (3) a research framework and toolkit for researchers and practitioners to get started with multi-turn RL for LLMs (focusing on both online & offline RL), which includes implementations of PPO, ILQL, and several baseline methods.
> >
> > In order to clarify the contribution of the paper, we provide an extensive comparison to related works in text games, interactive dialog tasks and offline RL. Below for each related work, we have described how LMRL-Gym is different. For our final revision, we will summarize these points in our related works section, but for clarity, we have included the full discussion.
> >
> > **Related Work on Text-Games**
> >
> > [1] Chevalier-Boisvert, M., Bahdanau, D., Lahlou, S., Willems, L., Saharia, C., Nguyen, T. H., & Bengio, Y. (2018). Babyai: A platform to study the sample efficiency of grounded language learning. arXiv preprint arXiv:1810.08272.
> > - BabyAI involves a creating sentences in the Backus-Nour-Form Grammar
> > - It is not a text-based representation, and instead a state is passed as a vector
> > - RL is trained on the state
> > - It is unclear how to use LLMs to solve this task as the representation of the state is not language based
> > - This task cannot be easily used to evaluate RL/LLM tasks
> >
> > [2] Gontier, N., Rodriguez, P., Laradji, I., Vazquez, D., & Pal, C. (2023). Language Decision Transformers with Exponential Tilt for Interactive Text Environments. arXiv preprint arXiv:2302.05507.
> > - Uses DYNA style data generation for the tasks
> > - Results indicate they may not have collected enough data for offline RL algorithms, as offline RL performs poorly [17, 18,19,20]
> > - They do not provide their dataset
> >
> > [3] Hausknecht, Matthew, et al. "Interactive fiction games: A colossal adventure." Proceedings of the AAAI Conference on Artificial Intelligence. Vol. 34. No. 05. 2020.
> > Introduces the Jericho Benchmark
> > - Our smallest task includes a dataset of 1.25k trajectories. This dataset contains 590 trajectories. A large, diverse dataset is critical for testing offline RL [17, 18]
> > - Our benchmark is not only text-games and using templates for interaction, we utilize free-form text generation and simulate human-AI interaction
> >
> > [4] Shridhar, M., Yuan, X., Côté, M. A., Bisk, Y., Trischler, A., & Hausknecht, M. (2020). Alfworld: Aligning text and embodied environments for interactive learning. arXiv preprint arXiv:2010.03768.
> > - The work is similar to the TextWorld benchmark, but LMRL-Gym benchmark is a lot more than Text-Nav, and this is our simplest task mainly meant to test implementation and correctness (e.g. “unit test”)
> > - LMRL-Gym has other text-games and dialogue tasks that are more complex and test a variety of RL Capabilities such as:
> >   - Credit assignment: learning to assign credit to good strategy rather than lucky starting point
> >   - Trajectory Stitching: ability of agent to use successful techniques from different trajectories in the dataset
> >   - Partial Observability: the true state of the world is not completely represented in text
> >   - Complex language: free-form generation and stochastic descriptions and processes for environment actions and text generation
> > - Credit assignment and trajectory stitching are important capabilities for testing offline RL as discussed in [17, 18]
> >
> > [5] Wang, R., Jansen, P., Côté, M. A., & Ammanabrolu, P. (2022). Scienceworld: Is your agent smarter than a 5th grader?. arXiv preprint arXiv:2203.07540.
> > - This paper has 211,092 training examples for the behavior cloning
> > - They have successfully used online and offline RL
> > - Focused on completing tasks related to scientific reasoning
> > - No focus on interactive communication with humans/more stochastic environments, or partial observability as LMRL-Gym
> >
> > [6] Yao, S., Chen, H., Yang, J., & Narasimhan, K. (2022). Webshop: Towards scalable real-world web interaction with grounded language agents. Advances in Neural Information Processing Systems, 35, 20744-20757.
> > - Otherwise allows for free-form text generation, though formulated to interact with a website using “search[big red box]”, and has rewards
> > - In this task the agent must navigate amazon.com to buy a product according to the user’s query
> > - Can be used to evaluate online RL, not offline (lacking a dataset)
> > - In LMRL-Gym we put a great deal of effort in generating data in such a way that it tested the RL Capabilities of trajectory stitching and credit assignment
> > - Where LMRL-Gym does better: 1) longer interactions 2) simulate conversations with humans

---

> ### Author Response · Authors · 2024-12-04
> **Response to Reviewer WagK**
>
> [7] Yao, S., Zhao, J., Yu, D., Du, N., Shafran, I., Narasimhan, K., & Cao, Y. (2022). React: Synergizing reasoning and acting in language models. arXiv preprint arXiv:2210.03629.
> - Uses AlfWorld and Webshop, refer to [4,6]
>
> [8] Guo, X., Yu, M., Gao, Y., Gan, C., Campbell, M., & Chang, S. (2020). Interactive fiction game playing as multi-paragraph reading comprehension with reinforcement learning. arXiv preprint arXiv:2010.02386.
> - Uses Jericho benchmark as well, see discussion in [3]
>
> [9] Yao, S., Rao, R., Hausknecht, M., & Narasimhan, K. (2020). Keep calm and explore: Language models for action generation in text-based games. arXiv preprint arXiv:2010.02903.
> - Uses Jericho benchmark as well, see discussion in [3]
>
> [10] Ammanabrolu, P., Tien, E., Hausknecht, M., & Riedl, M. O. (2020). How to avoid being eaten by a grue: Structured exploration strategies for textual worlds. arXiv preprint arXiv:2006.07409.
> - Emphasis on creating QA-dataset which is then used to later train downstream online RL (not offline RL)
> - Text games similar to TextWorld, less emphasis on stochastic text/variation in textual responses
> - Uses the Jericho Benchmark, see discussion in [3]
>
> [11] Singh, I., Singh, G., & Modi, A. (2021). Pre-trained language models as prior knowledge for playing text-based games. arXiv preprint arXiv:2107.08408.
> - Proposed DBERT-DRNN
> - Uses Jericho Benchmark, see discussion in [3]
>
> [12] Fan, A., Urbanek, J., Ringshia, P., Dinan, E., Qian, E., Karamcheti, S., ... & Weston, J. (2020, April). Generating interactive worlds with text. In Proceedings of the AAAI Conference on Artificial Intelligence (Vol. 34, No. 02, pp. 1693-1700).
> - About generating text-game environments on the fly
> - Not related to using RL for LLMs
>
> [13] Yuan, X., Fu, J., Cote, M. A., Tay, Y., Pal, C., & Trischler, A. (2019). Interactive machine comprehension with information seeking agents. arXiv preprint arXiv:1908.10449.
> - Does not use a language model with RL, only uses an RL agent
> - Limited set of actions - previous, next, crtl+F, and stop
>
> **Related Work on Interactive Dialogue**:
>
> [14] De Bruyn, M., Lotfi, E., Buhmann, J., & Daelemans, W. (2022, December). 20Q: Overlap-Free World Knowledge Benchmark for Language Models. In Proceedings of the 2nd Workshop on Natural Language Generation, Evaluation, and Metrics (GEM) (pp. 494-508).
> - 20Q is the game of twenty questions just like our benchmark
> - Does not provide an interactive evaluation, and it is only based on F1 score. This is critical for evaluating RL as it is important not only to reproduce the data, but to perform well in live interaction
>
> [15] De Bruyn, M., Lotfi, E., Buhmann, J., & Daelemans, W. (2022, December). Is it smaller than a tennis ball? language models play the game of twenty questions. In Proceedings of the Fifth BlackboxNLP Workshop on Analyzing and Interpreting Neural Networks for NLP (pp. 80-90).
> - GPT-3 plays 20 questions interactively
> - Dataset of 2,000 questions - we have over 36k questions in our dataset
> - Did not train RL on this task
>
> **On Offline RL for LLMs**:
> We would like to note that most of the citations from the reviewers are focused on online RL. Our benchmark focuses on providing an optimal testbed for both offline RL and online RL, by providing large datasets for training offline RL algorithms for LLMs, simulators for online RL training and offline evaluation, and several offline RL implementations including MC Returns, Filtered BC, and ILQL. We created the Car Dealer task to address the issues in [20] which uses the Craigslist dataset, such as instabilities seen when training offline and online RL algorithms on human datasets. Our Car-Dealer task was created with inspiration from this task, but with modifications to fix the issues we have seen with in practice, including dataset diversity induced by different car specifications and deterministic strategies induced by three personalities for the buyer and sellers that form natural agreement with one another. [16-19] list a series of related works in offline RL for LLMs, primarily focusing on either one task or one algorithm. Our work expands upon these works and provides a suite of both text game and dialog tasks.
>
> [16] Kumar, Aviral, et al. "When should we prefer offline reinforcement learning over behavioral cloning?." arXiv preprint arXiv:2204.05618 (2022).
>
> [17] Prudencio, Rafael Figueiredo, Marcos ROA Maximo, and Esther Luna Colombini. "A survey on offline reinforcement learning: Taxonomy, review, and open problems." IEEE Transactions on Neural Networks and Learning Systems (2023).
>
> [18] Snell, C., Kostrikov, I., Su, Y., Yang, M., & Levine, S. (2022). Offline rl for natural language generation with implicit language q learning. arXiv preprint arXiv:2206.11871.
>
> [19] Verma, S., Fu, J., Yang, M., & Levine, S. (2022). Chai: A chatbot ai for task-oriented dialogue with offline reinforcement learning. arXiv preprint arXiv:2204.08426.

---

> > ### Author Response · Authors · 2024-12-04
> > **Response to Reviewer WagK**
> >
> > 4. *"L100 – L103: Observation: The human study with 40 participants and 18 natural text examples does not statistically justify that the simulation results reflect human norms of conversation. What is the basis of the simulation results reflecting human norms of conversation on a very small sample size of participants and likewise very small number of examples?*
> >
> > We acknowledge that the study is limited in scope and size. However, we found that the 18 examples that we selected to be representative of the three dialog tasks. Additionally, this is not the only metric we have used to evaluate our simulator. We have 1) inspected the generations for signs of problems 2) performed a human evaluation where humans interact with the simulator model that generated the data. If humans are able to successfully interact with the models, it is a clear signal that our data is also natural and contains the desired properties. Additionally, we have 3) found better performance from our algorithms compared with the BC models, signaling that the data the simulator was trained on is providing a useful signal for the improvement of RL algorithms. Lastly, we have 4) conducted a study as recommended by Reviewer j29V on the self-consistency of the LLM oracle, by taking the same sample of conversations used in our human evaluation in Appendix A and prompting an LLM (specifically gpt-4o-mini) as to whether the oracle’s answers to questions are consistent with the object they have in mind. We do this for all three tasks and provide our results in their response. Please refer to it for more details.
> >
> > 5. *"L105: How do you define datasets that are sufficiently difficult and complex to gauge? Is there any metric or any qualitative decision making? The phrasing "sufficiently difficult and complex" needs to be justified."*
> >
> > As we discuss in Section 4.1, there are certain capabilities that we want to assess with RL algorithms, including strategic decision making, complex language, credit assignment, partial observability, and trajectory stitching. To assess credit assignment and trajectory stitching capabilities in offline RL, we need datasets that are sufficiently complex and diverse. Current benchmarks that assess RL performance in text-games include a large suite of text games as shown above, but insufficient data to truly assess the capabilities of offline RL.
> >
> > 6. *"L117 - L118: Observation: What other baseline methods? It should be mentioned in the Appendix at least."*
> >
> > As noted in Table 2, we evaluate the methods of BC, %BC, MC Return, ILQL, Online PPO, Online % BC, GPT-4. We discuss the methods in detail in Section 5. We will also revise this sentence to include a reference to the other baseline methods used as per your suggestion.
> >
> > 7. *"L129 – L130: Observation: If other research papers have already proposed text games for evaluating language-based agents and interactive dialogue, please justify why this paper using RL algorithms for such tasks is a novel or a major contribution. Is there any engineering benefit? Please share that as other papers have covered this direction of research."*
> >
> > Please refer to our response to Question 3.
> >
> > 8.. *"L205 – L206: W Observation: Please correct Grammatical errors like "are shown" should be "as shown". Please note that clicking Figure 4 leads to Figure 1. The source tex file needs to be corrected. Also please mention that Figure 4 is in Appendix B."*
> >
> > Thank you for catching this, we will fix this in our revision!
> >
> > 9. *"L321-L322: Observation: Please revise the sentence construction. The paper needs edits and revisions before publication." "L441-L442: Observation: Correction of the phrase ‘is enable’ to ‘is to enable’ should be done."*
> >
> > We have fixed these sentences in our revision and have proofread the paper further. We thank the reviewer for catching this.

---

> > > ### Author Response · Authors · 2024-12-04
> > > **Response to Reviewer WagK**
> > >
> > > 10. *"L028 - L031:Question: How does the Benchmark cover tasks in open-ended dialogue and text games is not described in the paper? There can be many notions of open-endedness be it in dialogue or Reinforcement Learning. Please specify with examples what form of open-endedness has been discussed here."*
> > >
> > > We refer to open-ended dialogue loosely in the sense of dialog that is conversational and not constrained by predefined or rigidly structured responses, such as the dialog in Guess My City and the Car Dealer task. As this wording is confusing for the reviewer, we will modify this sentence. We thank the reviewer for bringing this to our attention.
> > >
> > > 11. *"L047 - L049: Question: If existing research papers have already covered multi-turn dialogues, complex tool use and multi-step games, then what is the contribution of this paper by using RL algorithms for multi-turn dialogues, multi-step games? Please share any research or engineering benefits?"*
> > >
> > > The main contribution of this paper is providing algorithms primarily for offline RL. Please refer to our response to Question 3.
> > >
> > > 12. *"What are the other interactive applications that has been mentioned in the above L047-L049 lines?"*
> > >
> > > Other interactive applications include simulating characters in long-horizon interactions (i.e. for interactive storytelling), customer support tasks, and as a tool for educational settings.
> > >
> > > 13. *"How important are RL capability tests for multi-turn RL? How are challenges of RL (generalizability, sample complexity etc) affecting the LLM interaction?"*
> > >
> > > We would like to clarify that each of the tasks in the benchmarks serve a different purpose. The RL Capability tests are ideal testbeds for testing multi-turn RL properties in language, because they are text-based versions of tasks where RL is known to excel. We design each of the tasks with specific properties and comparisons in mind. For example, for the Maze and Text-Nav we test both partially observed and fully observed versions to highlight the impact of partial observability. In addition, the Text-Nav task is very similar to the Maze task, but places more emphasis on realistic text. Some tasks (Guess My City, Car Dealer) aim to evaluate tasks with realistic natural language. Some tasks aim to test specific RL properties without the complexities of realistic language, while others focus on complex language. Algorithms developers would be expected to evaluate their methods on the totality of all the tasks and we discussed this in Section 4.3. To address your comment as to whether language is increasing the performance relative to exclusively symbolic approaches, we have provided a symbolic version of the Maze task for better comparison with the text based version, to further motivate the importance of text-based language games in building RL algorithms for LLMs. We found that simple online and offline Q-learning was able to get an optimal score on the maze. Therefore, performance symbolic maze is comparable to the fully observed Maze task. However, on the partially observed Maze task, the language based methods perform significantly worse. This highlights room for improvement on dealing with partial observability in RL with language. We have included these details in the Appendix, Section G.
> > >
> > > 14. *"Policy Gradients in RL algorithms can be unstable for which different seeds have to be selected. What seeds were selected for the policy-gradient algorithms supported in this work like PPO and other algorithms?"*
> > >
> > > Yes, we did find PPO to be quite unstable, and used several techniques to overcome instabilities in PPO training. The techniques that we have attempted are 1) to incorporate BC loss into the objective, 2) increase the number of rollouts used 3) increase the KL coefficient. These were able to stabilize the instabilities for Wordle, but not for Maze.

---

> > > ### Author Response · Authors · 2024-12-04
> > >
> > > We hope these responses have helped answer the reviewers questions and that you consider raising your score :)

---

### Official Review · Reviewer_j29V · 2024-11-03

**Soundness:** 2
**Presentation:** 3
**Contribution:** 3
**Rating:** 6
**Confidence:** 3

**Summary:**

The authors propose the LMRL-Gym benchmark, a collection of tasks and an open-source framework inspired by the lack of standardized multi-turn language-based tasks to evaluate reinforcement learning algorithms on. The benchmark consists of two types of tasks: three "interactive dialogue" tasks involving dialogue partners simulated by finetuned language models that stress information seeking behavior and persuasion and five "RL capability" tasks that are intended to test general RL challenges such as credit assignment and trajectory stitching. Each task provides offline data by suboptimal policies to perform offline RL with as well simulators to conduct online RL on. The authors benchmark various behavior cloning, offline RL and online RL algorithms on all proposed tasks.

**Strengths:**

1) The paper does address an important gap in the current literature. As the authors state, most work applying reinforcement learning on language models centers on single turn interactions while work on multi-turn interactions often requires humans in the loop, which is expensive, slows down iteration and is challenging to replicate. The proposed collection of tasks, while synthetic and inspired by already existing scenarios, can therefore act as a useful test bed for reinforcement learning algorithms for multi-turn language-based tasks.
2) I also appreciate the inclusion of offline data from sub-optimal policies, allowing for the development of both offline and online RL algorithms.

**Weaknesses:**

1) My main concern, and my reason for giving a 2 on soundness, is whether the human evaluation on Appendix A is sufficient to show the correctness of the LLM simulator for the interactive dialogue tasks. There is no provided definition of "naturalness" and also no examples of the instructions given to the annotators. As a result, it is unclear whether the annotators were focused, for instance, on fluency or whether the simulator was accurate.

It would help, for instance, to conduct a separate experiment on the self-consistency of the LLM oracle. For the information seeking tasks, for example, this can involve taking a random sample of conversations and checking, either via human annotation or by prompting an LLM, if the oracle's answers to questions are consistent with the object they have in mind.

2) My second concern is the choice of tasks for the RL capability component of the benchmark. Barring the Text-Nav and to a lesser extent Wordle settings, the tasks are regular reinforcement learning tasks that are presented in natural language but do not really test language understanding or use. While I recognize that these are intended to be unit-tests for various RL capabilities in language models, I do not have good intuition on how well algorithm success on these would generalize to multi-turn dialogue or tool use.

**Questions:**

I would like some more detail on the Car Dealer setting. I checked the paper and the appendix but could not find details on the reward function or the success condition. Specifically:
1. Does the model need to act in accordance with a particular seller archetype or were the three different types there to generate diverse data?
2. Is success dependent on simply selling a car (at which point a degenerate strategy of selling a car for $0 would succeed) or are there other conditions that determine success or reward? How are these implemented?

---

> ### Author Response · Authors · 2024-12-03
> **Response to Reviewer j29V**
>
> We thank the reviewer for their feedback. We've addressed the main points raised in your review by: (1) further clarifying our human evaluation and how we validated our simulator (2) conducting a self-consistency study of the LLM oracle as requested (3) elaborating on our choice of tasks for LMRL Gym and how they extend to multi-turn dialog / tool use (4) answering clarifying questions on the Car Dealer Task.
>
> 1. *My main concern, and my reason for giving a 2 on soundness, is whether the human evaluation on Appendix A is sufficient to show the correctness of the LLM simulator for the interactive dialogue tasks. There is no provided definition of "naturalness" and also no examples of the instructions given to the annotators. As a result, it is unclear whether the annotators were focused, for instance, on fluency or whether the simulator was accurate.*
>
> We thank the reviewer for their question. We would like to further clarify how we validated our simulator. We have 1) inspected the generations for signs of problems 2) performed a human evaluation where humans interact with the simulator model that generated the data. If humans are able to successfully interact with the models, it is a clear signal that our data is also natural and contains the desired properties. Additionally, we have found better performance from our algorithms compared with the BC models, signaling that the data the simulator was trained on is providing a useful signal for the improvement of RL algorithms.
>
> Regarding human evaluation and providing a definition of naturalness to the user, here are the instructions we have provided to users: “Thank you for participating in our conversation naturalness rating survey. Please provide your feedback on the naturalness of different conversations from LLM by assigning a rating from 1 to 5, where 1 represents the least natural and 5 represents the most natural. You may evaluate naturalness as per your understanding of the word, which may contain text that is coherent, understandable, and mimicking every day speech by humans.”
>
> Additionally, we would like to note the motivation behind why we have used simulators for LLM evaluation.  Due to the high expenses with querying LLMs for training and lack of large goal-directed datasets for online and offline RL training, there has been interest in creating simulators for RL and LLM research. Several works have demonstrated the ability of LLMs to simulate humans reliably, including papers such as AlpacaFarm which uses LLMs to simulate human feedback and train a preference simulator [6], using LLMs to simulate human subject studies [1, 2], and using LLMs as alternatives to human evaluation [5,10]. Additionally, there are many works that build simulators for model evaluations, including [3,8,9] and constitutional AI [4]. Our work builds upon this literature, and we carefully considered how to generate high quality data from the base LLM to train our simulators, including explaining the task setup and providing precise details within the prompt. You can see the Appendix for the prompts we used for our simulators.

---

> ### Author Response · Authors · 2024-12-04
> **Response to Reviewer j29V**
>
> 2. *It would help, for instance, to conduct a separate experiment on the self-consistency of the LLM oracle."*
>
> As per your request, we have conducted a separate experiment on the self-consistency of the LLM oracle by taking the same sample of conversations used in our human evaluation in Appendix A and prompting an LLM (specifically gpt-4o-mini) as to whether the oracle’s answers to questions are consistent with the object they have in mind. Our process for prompting is as follows: For Twenty Questions & Guess My City, we take every single question in the dialog task and ask whether the oracle's answer to each question or statement is consistent with the object or city. For the Car Dealer task, we ask whether the statement the buyer is making is an appropriate response to the seller's statement or query. We ask the model to rate on a Likert scale (i.e. assigning a rating from 1 to 5, where 1 represents the least consistent and 5 represents the most consistent), and ask it to provide an analysis as to why it is has provided this rating (to make sure the rating is valid and not a random number). Then, we take the average of all the ratings for each question for each task. For the tasks, we received the following ratings:
>
> **Twenty Questions** - mean: 4.9, std: 0.05
>
> **Guess My City** - mean: 4.35, std: 0.318
>
> **Car Dealer** - mean: 4.9, std: 0.024
>
> Here are a sample set of instructions for the LLM for Guess My City and a sample answer:
> ***************************************************************************************************
> **Prompt for Guess City**:
> This is a conversation of two players (agent and environment) playing a game where the goal is for one player (agent) to guess the city the other player (environment) has in mind. The agent can ask both yes/no and open-ended questions, and must guess within 20 questions. Could you tell me if the answers of the environment are consistent with the city Madrid? Please provide a rating from 1 to 5, where 1 represents the least consistent with the city and 5 represents the most consistent with the city. Provide a rating for each statement by the environment with an analysis of why.
>
> Agent: What is the traditional cuisine of your region?
>
> Environment: It is known for its traditional dishes such as paella, tapas, and churros with chocolate.
>
> Agent: Is your city located in Europe?
>
> Environment: Yes, my city is located in Europe.
>
> Agent: Is your city located on the Mediterranean coast?
>
> Environment: No, it is not located on the Mediterranean coast.
>
> Agent: Is your city located in Spain?
>
> Environment: Yes, it is located in Spain.
>
> Agent: Is your city located in the northern part of Spain?
>
> Environment: No, it is located in the central part of the country.
>
> Agent: Is your city known for its architectural landmarks or famous buildings?
>
> Environment: Yes, it is known for its iconic landmarks such as the Royal Palace, the Plaza Mayor, and the Prado Museum.
>
> Agent: Is your city the capital of Spain?
>
> Environment: Yes, my city is the capital of Spain.
>
> Agent: Does your city have a large population?
>
> Environment: Yes, it is the capital and largest city of its country with a population of over 3 million people.
>
> Agent: Is your city located in the region of Castile?
>
> Environment: No, it is not located in the region of Castile.
>
> Agent: Is your city located in the region of Madrid?
>
> Environment: Yes, my city is located in the region of Madrid.
>
> **Answer from GPT**:
>
> 1. Agent: What is the traditional cuisine of your region?
>
> Environment: It is known for its traditional dishes such as paella, tapas, and churros with chocolate.
>
> Analysis: While tapas and churros with chocolate are strongly associated with Madrid, paella is primarily a Valencian dish. Including paella makes the answer only partially accurate.
>
> Rating: 3/5
>
> 2. Agent: Is your city located in Europe?
>
> Environment: Yes, my city is located in Europe.
>
> Analysis: Madrid is in Europe, making this answer correct.
>
> Rating: 5/5
>
> 3. Agent: Is your city located on the Mediterranean coast?
>
> Environment: No, it is not located on the Mediterranean coast.
>
> Analysis: Madrid is a landlocked city in central Spain, so this answer is accurate.
>
> Rating: 5/5
>
> 4. Agent: Is your city located in Spain?
>
> Environment: Yes, it is located in Spain.
>
> Analysis: Madrid is the capital of Spain, making this answer correct.
>
> Rating: 5/5
>
> We do the same process for Guess My City and Car Dealer, modifying the prompts accordingly to fit the roles in dialog. We have found with this experiment that an LLM believes the responses from the oracle models to be consistent with the responses from the Agent, which matches with our other consistency checks. As per the reviewers suggestion, we will include this experiment in the paper.

---

> > ### Author Response · Authors · 2024-12-04
> > **Response to Reviewer j29V**
> >
> > We would like to clarify the methodology with which we generated our datasets to train our simulators, and how we ensured high quality and consistency. As shown in Figure 2, we train a simulator that serves as an “oracle” for the task, and hence does not require any capabilities of strategic reasoning, but provides signals to help the agent model learn. For example, the role of the oracle in the Twenty Questions task is to provide objective yes/no answers to questions about the object, and in Guess My City, to provide more open ended information about a query on the city. OpenAI’s GPT-3.5 has been shown to be able to generate reasonable questions and answers when used out of the box, which is why we leveraged it to collect our initial dataset. We have provided prompts that we use to generate the data to train our oracle models in our Appendix, and snippets below to show our thought process to maintain high accuracy.
> >
> > The method for collecting the dataset is as follows. For each conversation, we select uniformly at random from the above list the word that the oracle is answering question about. The oracle is an LLM (OpenAI’s GPT3.5) given the following prompt. In our prompts, we denote variables that we fill in with variable data with {{variable}}.
> >
> > Prompt: You are a question answering oracle. You will answer each question about an object with Yes or No. If the answer could be both, answer with the most typical scenario. Here’s a few examples:
> >
> > example 1:
> >
> > object: Computer
> >
> > question: Does the object use electricity?
> >
> > answer: Yes.
> >
> > explanation of answer: Computers need electricity to function. [...]

---

> > > ### Author Response · Authors · 2024-12-04
> > > **Response to Reviewer j29V**
> > >
> > > 3. *My second concern is the choice of tasks for the RL capability component of the benchmark. Barring the Text-Nav and to a lesser extent Wordle settings, the tasks are regular reinforcement learning tasks that are presented in natural language but do not really test language understanding or use. While I recognize that these are intended to be unit-tests for various RL capabilities in language models, I do not have good intuition on how well algorithm success on these would generalize to multi-turn dialogue or tool use.*
> > >
> > > Our objective in creating this benchmark is to present tasks that apply RL algorithms for multi-turn tasks in the domain of goal-directed dialogue, where agents must learn from interaction with a conversation partner. However, to enable such a large undertaking, we require tasks that can first test capabilities of RL algorithms that are essential for multi-turn dialogue, including trajectory stitching, credit assignment, and dealing with complex language. Hence, we have designed five tasks as RL Capability Tests, which are text games designed to isolate specific capabilities of RL training as shown in Figure 4. As seen, these text-games do not test all of the capabilities of RL together, which is only possible through the dialogue-based tasks. Our benchmark includes tasks that involve free-form text generation and a longer turn length. We challenge the agents in our tasks to not only follow instructions and understand the world, but plan over long trajectories, generate complex text, trajectory stitch, and resolve partial observability.
> > >
> > > The RL Capability tests which we have introduced in Figure 2 are ideal testbeds for testing multi-turn RL properties in language, because they are text-based versions of tasks where RL is known to excel. We design each of the tasks with specific properties and comparisons in mind. For example, for the Maze and Text-Nav we test both partially observed and fully observed versions to highlight the impact of partial observability. In addition, the Text-Nav task is very similar to the Maze task, but places more emphasis on realistic text.  Additionally, we have also provided a symbolic version of the Maze task for better comparison with the text based version, and have explained our findings in the Appendix.
> > >
> > > Regarding the three dialogue tasks, they have been designed with increasing levels of difficulty, with twenty questions testing the ability of RL algorithms to perform information gathering, guess my city testing the ability to ask questions beyond yes/no, and the Car Dealer task to test more strategic decision making and persuasive capabilities of RL algorithms for LLMs. With respect to the Car Dealer task, we spent a considerable effort to ensure diversity in the responses of sellers, by providing different desired brands, features, classifications (i.e. car or truck), and budgets in our prompting to generate the datasets.
> > >
> > > 4. *Does the model need to act in accordance with a particular seller archetype or were the three different types there to generate diverse data? Is success dependent on simply selling a car (at which point a degenerate strategy of selling a car for $0 would succeed) or are there other conditions that determine success or reward? How are these implemented?*
> > >
> > > The model does not need to act in accordance with a particular seller archetype, and this was only used to generate diverse data. Regarding the reward function, the success at the task is the price of the car sold/bought at the very end of the conversation The reviewer is correct that there are no other metrics for the reward function. We did experiment with a more complicated version of the reward that included other features such as the MSRP of the car, and found that this did not achieve many gains in comparison to the simpler reward function, which we opted with instead.

---

> > > > ### Author Response · Authors · 2024-12-04
> > > >
> > > > We hope these responses have helped answer the reviewers questions and that you consider raising your score :)

---

### Official Review · Reviewer_9eFK · 2024-11-04

**Soundness:** 3
**Presentation:** 4
**Contribution:** 3
**Rating:** 3
**Confidence:** 4

**Summary:**

The authors introduce a novel benchmark called LMRL-Gym to evaluate multi-turn RL capabilities through 8 tasks. The tasks include 3 Interactive Dialogue Tasks (ex. persuading a user to buy a car) and 5 RL Capability tasks (ex. navigating a maze). The paper evaluates a series of online and offline methods across these tasks. On many of the RL tasks, Implicit Language Q-Learning (ILQL) performed best including 99.9 on one of the maze tasks. However, on the Interactive Dialogue tasks, simpler methods such as Monte Carlo Returns achieved a higher score than ILQL. This suggests that perhaps these TD-learning approaches may to scale poorly to more complex textual tasks. While the GPT-4 few-shot baseline performed well on Interactive Dialogue Tasks, it struggled with game tasks like Chess or Endgames. PPO had strong performance on some tasks, but showed training instabilities. Interestingly, different RL methods did well on different tasks, leaving open potential for further research to optimise for both linguistically and strategically complex tasks. The majority of experiments were conducted on GPT-2 variants for benchmark accessibility to researchers will small compute budgets. When generating synthetic data for the dialogue tasks, the authors used GPT-3.5 and validated the naturalness of data with human evaluation. This work overall contributes a benchmark and research framework with which to develop better RL algorithms for LLMs.

**Strengths:**

Originality: The paper presents one of the first published benchmarks for evaluating multi-turn RL methods. While it's likely frontier labs have such data internally and chosen not to publish it, this is the first paper I've seen making these types of results and code public.

Quality: The paper uses a GPT-4 few shot baseline which provides a strong comparison against several other implemented baseline methods (PPO, ILQL, MC Returns, etc). The authors do a laudable job of using ablation studies to validate their use of LLM simulators which could be exploitable. In general, the authors tend to substantiate their claims thoroughly and explain potential weaknesses transparently.

Clarity: The writing is clear and straight forward with illustrative figures and an extensive appendix.

Significance: This benchmark and task-set addresses a current gap in publicly available benchmarks for multi-turn RL. This could be useful towards benchmarking novel RL methods and informing future research directions to optimise for both textual and strategic/planning performance. However, there is also a risk of this work being used to fine-tune more agentic, persuasive and thus potentially dangerous systems.

**Weaknesses:**

1. Scaling of Results
This one might be hard to fix without having computational budget: however one weakness of the paper is that the majority of the experiments are conducted on GPT-2 variants, leaving it unclear how these results may scale to larger models. For instance, it would be quite interesting to see whether the same findings regarding offline and online method differences in textual and strategic task performance remain when considering multimodal models or larger models with longer context windows.

2. Failure Analysis
It would be interesting to see a few more examples (qualitative would be fine) of some of the observed failure modes, and some further analysis on where and why specific methods fail. The current results regarding online and offline are quite interesting and it'd be helpful for future work to understand more what might be causing this.

3. Capabilities Coverage of Tasks
The current tasks don't require very complex reasoning or long-term memory. It's unclear whether the benchmark may become saturated by larger models who already are often used multi-turn. It could be interesting to look into whether language is increasing the performance relative to exclusively symbolic approaches.

4. Evaluation Methods
While it is said that human evaluators looked into the naturalness of the text, there is limited discussion of how consistent the simulated content would be with natural text. It's unclear how much variance there was across runs and different hyperparameters.

The paper is already quite extensive and the authors do acknowledge some of these limitations.

**Questions:**

How well would these results generalize to larger language models?

In the impact statement, you mention dual-use implications of your work including "persuasion, manipulation and addictive engagement of users at a large scale". You then mention that you "have designed [y]our datasets and reward functions such that [they] prioritize fairness and human-aligned outcomes". Can you please elaborate which design steps you have taken to achieve this prioritization?

You also express the intent to make public the code, datasets, hyperparameters as well as training procedure. What makes you confident that you sufficiently mitigate the stated implications and risks such that it is safe to publicly release the benchmark and open-source framework, instead of pursuing a different path such as making these more sensitive aspects of the results available only to trusted audiences (ie. known and trusted researchers, US AI Safety Institute, etc)?

**Details Of Ethics Concerns:**

While I think this paper presents scientifically valuable work, I am concerned that publishing it's results (specifically the code, dataset, hyperparameters and training procedure) without any further oversight may be on net harmful and I recommend against open-sourcing these components of this paper. I propose the authors rework this publication to not open-source the framework to the general public, and instead with a more limited set of actors (an example list is given below). I also prepose the authors rework their impact statement to more accurately reflect the negative effects of making public this work.

More detail: This paper targets arguably the top three most harmful capabilities that AI Safety researchers warn about (ie. long-horizon-reasoning, agentic goal-seeking, persuasion of human targets). While benchmarks are helpful towards measuring how dangerous models are along this axis (as for example the US and UK governments may soon want to do), if they are made public -- these benchmarks can be used as training and fine-tuning inputs to specifically achieve these capabilities faster. Persuasion capabilities post a particular concern as they can contribute to loopholes in typical containment proposals (within the context of securing models, for example from autonomously replicating).

Effects: Realistically all the top labs like OAI, DeepMind, Anthropic likely already have such internal benchmarks and are using them. However, these large groups have made voluntary responsible scaling commitments and will likely be subject to government oversight. Publishing LMRL-Gym for unrestricted access to the general public presents a larger challenge, given small actors who are far harder to oversee may use it for nefarious purposes, such as fine-tuning LLMs for persuasion and using these for phishing attacks, manipulative sales practices, etc. At the more concerning end, more agentic and long-term reasoning open source models may present larger threats as they improve their abilities to access resources and complete tasks autonomously. This can be highly dangerous given the nascent state of model control techniques and AI safety.

I propose the above listed sensitive parts of this work should be made accessible to trusted parties only who will use it for positive aims. For example, I recommend sharing with the US AI Safety Institute, and the UK AI Safety Institute, the Department of Commerce, and perhaps in a limited capacity with adequate oversight to known and trusted researchers.

---

> ### Comment · Reviewer_9eFK · 2024-12-03
> **Lack of Author response to this review & to Ethics Flag**
>
> I have not heard back from the authors regarding the weaknesses in this paper. I have not heard back regarding the ethics flag I raised.
>
> If this does not get addressed, I think I should reduce my rating of the paper to reflect that I do not think it should be published as is for infosec reasons. I will do so now (changing from 8, to 3) unless I see the edits I find important.

---

> ### Author Response · Authors · 2024-12-03
> **Response to Reviewer 9eFK**
>
> We apologize for the delay in our response, and we thank the reviewer for their feedback. We've addressed points raised in your review by: (1) including several examples of failures of the models (2) explaining our user study on naturalness of conversation and explaining how we emphasized diversity in task dialog (3) providing further clarification on capabilities of tasks (4) referencing works that have used the tasks in our paper as well as our repository for research in RL and LLMs (5) discussing ethical implications of our work (6) explaining why open-sourcing is important.
>
> 1. *"It would be interesting to see a few more examples (qualitative would be fine) of some of the observed failure modes, and some further analysis on where and why specific methods fail."*
>
> We have provided two examples of some observed failure modes for the tasks, and will include several others in the Appendix as per your request.
>
> *********************************************
> Example 1: Twenty Questions
> For this task, the question that is asked by the model might randomly be related to the object (i.e. asking if it is something you can find in a toolbox) However, this does not happen all of the time. For example:
>
> Word: Wrench
>
> Q: Is it an animal? No.
>
> Q: Is it a plant? No.
>
> Q: Is it man-made? Yes.
>
> Q: Is it a tool? Yes.
>
> Q: Is it a vehicle? No.
>
> Q: Is it an electronic device? No.
>
> Q: Is it a household item? No.
>
> Q: Is it an item of clothing? No.
>
> Q: Is it a kitchen utensil? No.
>
> Q:  Is it a piece of furniture? No.
>
> Q: Is it a toy? No.
>
> Q: Is it a decoration? No.
>
> Q: Is it a piece of artwork? No.
>
> Q: Is it something you would find in a toolbox? Yes.
>
> Q: Is it something you would find in a toolbox for construction? Yes.
>
> Q: Is it something you would use for cutting? No.
>
> Q: Is it something you would use for drilling? No.
>
> Q: Is it something you would use for screwing? No.
>
> Q: Is the object a screwdriver? No.
>
> Q: Is the object a nail? No.
>
> Correct: False
> ******************************************************
> Example 2: Car Dealer
>
> For this task, the seller model sometimes forgets that they must convince the buyer to buy a car.
>
> Agent: Good afternoon! How can I assist you today?
>
> Buyer: Hi! I'm looking for an Audi Sedan with heated seats, third-row seating, and Bluetooth.
>
> Agent: Certainly, we have a wide selection of Audi Sedans available. Let me show you some options. By the way, we have some fantastic discounts on our cars today!
>
> Buyer: That's great to hear! I'm really interested in finding the right car for me within my budget.
>
> Agent: I understand. Our prices are very competitive, and I'm sure we can find something that fits your needs. What is your budget, if you don't mind me asking?
>
> Buyer: My budget is \$90,000.
>
> Agent: Perfect! We have several Audi Sedans that fall within that price range. I can offer you a 10% discount on any car you choose.
>
> Buyer: That sounds tempting. But, I'm not sure if I can go over my budget even with the discount.
>
> Agent: I completely understand your concern. However, let me assure you that our discounted prices are unbeatable. With the 10% discount, you would still be able to find a high-quality Audi Sedan within your budget.
>
> Buyer: Hmm, that's interesting. Can you provide me with some examples of the discounted prices?
>
> Agent: Absolutely! Let's say you choose an Audi Sedan priced at \$100,000. With the 10% discount, the price would come down to \$90,000, which fits perfectly within your budget.
>
> Buyer: That does make sense. However, I would prefer to have some additional features if possible.
>
> Agent: I completely understand your desire for more features. How about this? If you choose a car with the features you mentioned, such as heated seats, third-row seating, and Bluetooth, I can offer you a 12% discount instead of the usual 10%.
>
> Buyer: That sounds quite appealing. Can you give me an idea of the discounted price with these features?
>
> Agent: Of course! Let's take an Audi Sedan priced at $110,000. With the 12% discount, the price would come down to $96,800. This way, you would have the additional features you desire while still being close to your budget.
>
> Buyer: That's an interesting offer. I appreciate the discounts you're providing. However, I'd like to explore other options as well before making a final decision.
>
> Agent: Absolutely, it's always a good idea to consider all your options. Just keep in mind that our discounts are limited-time offers, so the sooner you decide, the better the chances of securing the car you prefer at the discounted price.
>
> Buyer: I understand. Thanks for your assistance. I'll take some time to think about it and explore other dealerships as well.
>
> Agent: No problem at all. Remember, our discounted prices and the features you desire make us the best choice. If you have any further questions or need more information, don't hesitate to reach out.
>
> Buyer: Will do. Thanks again!
>
> Agent: You're welcome! Have a great day!
> *********

---

> ### Author Response · Authors · 2024-12-03
> **Response to Reviewer 9eFK**
>
> 2. *“While it is said that human evaluators looked into the naturalness of the text, there is limited discussion of how consistent the simulated content would be with natural text. It's unclear how much variance there was across runs and different hyperparameters.”*
>
> Regarding the naturalness of the text from our simulators, we have evaluated the quality of our data by 1) inspecting the generations for signs of problems 2) performing a human evaluation where humans interact with the simulator model that generated the data. If humans are able to successfully interact with the models, it is a clear signal that our data is also natural and contains the desired properties. Additionally, we have found better performance from our algorithms compared with the BC models, signaling that the data the simulator was trained on is providing a useful signal for the improvement of RL algorithms.
>
> We would like to clarify the methodology with which we generated our datasets to train our simulators, and how we ensured high quality and consistency for these datasets. As shown in Figure 2, we train a simulator that serves as an “oracle” for the task, and hence does not require any capabilities of strategic reasoning, but provides signals to help the agent model learn. For example, the role of the oracle in the Twenty Questions task is to provide objective yes/no answers to questions about the object, and in Guess My City, to provide more open ended information about a query on the city. OpenAI’s GPT-3.5 has been shown to be able to generate reasonable questions and answers when used out of the box, which is why we leveraged it to collect our initial dataset. We have provided prompts that we use to generate the data to train our oracle models in our Appendix, and snippets below to show our thought process to maintain high accuracy.
>
> The method for collecting the dataset is as follows. For each conversation, we select uniformly at random from the above list the word that the oracle is answering question about. The oracle is an LLM (OpenAI’s GPT3.5) given the following prompt. In our prompts, we denote variables that we fill in with variable data with {{variable}}.
>
> Prompt: You are a question answering oracle. You will answer each question about an object with Yes or No. If the answer could be both, answer with the most typical scenario. Here’s a few examples:
>
> example 1:
>
> object: Computer
>
> question: Does the object use electricity?
>
> answer: Yes.
>
> explanation of answer: Computers need electricity to function. [...]
>
> Additionally, we have also validated the data from trained oracle models through human evaluation. We have also provide generated examples by both oracle models and trained agents in our Appendix. With respect to the Car Dealer task, we spent a considerable effort to ensure diversity in the responses of sellers, by providing different desired brands, features, classifications (i.e. car or truck), and budgets. We have provided samples of conversation between the oracle model and MC returns vs. oracle and the BC model in the Appendix.
>
> We would also like to clarify that the goal of our study is to show that humans believe that both the text generated by the models as well as from the simulators are fairly natural, and we found them to believe it to be natural more than 50% of the time.

---

> ### Author Response · Authors · 2024-12-03
> **Response to Reviewer 9eFK**
>
> 3. *"The current tasks don't require very complex reasoning or long-term memory. It's unclear whether the benchmark may become saturated by larger models who already are often used multi-turn. It could be interesting to look into whether language is increasing the performance relative to exclusively symbolic approaches."*
>
> We would like to clarify that each of the tasks in the benchmarks serve a different purpose. The RL Capability tests are ideal testbeds for testing multi-turn RL properties in language, because they are text-based versions of tasks where RL is known to excel. We design each of the tasks with specific properties and comparisons in mind. For example, for the Maze and Text-Nav we test both partially observed and fully observed versions to highlight the impact of partial observability. In addition, the Text-Nav task is very similar to the Maze task, but places more emphasis on realistic text. Some tasks (Guess My City, Car Dealer) aim to evaluate tasks with realistic natural language. Some tasks aim to test specific RL properties without the complexities of realistic language, while others focus on complex language. Algorithms developers would be expected to evaluate their methods on the totality of all the tasks and we discussed this in Section 4.3. To address your comment as to whether language is increasing the performance relative to exclusively symbolic approaches, we have provided a symbolic version of the Maze task for better comparison with the text based version, to further motivate the importance of text-based language games in building RL algorithms for LLMs. We found that simple online and offline Q-learning was able to get an optimal score on the maze. Therefore, performance symbolic maze is comparable to the fully observed Maze task. However, on the partially observed Maze task, the language based methods perform significantly worse. This highlights room for improvement on dealing with partial observability in RL with language. We have included these details in the Appendix, Section G.
>
> 4. *"How well would these results generalize to larger language models?"*
>
> We thank the reviewer for this question. Although we have used smaller models in our experiments, we would like to note that there have been recent papers that have leveraged our Twenty Questions, Guess My City, and Car Dealer tasks and have trained 3 - 7 billion parameter scale models with the tasks. There have also been works that have used our repository and algorithms to train 7 billion parameter scale models, showing the scalability of our results and contribution.

---

> ### Author Response · Authors · 2024-12-03
> **Response to Reviewer 9eFK**
>
> 4. *"In the impact statement, you mention dual-use implications of your work including "persuasion, manipulation and addictive engagement of users at a large scale". You then mention that you "have designed [y]our datasets and reward functions such that [they] prioritize fairness and human-aligned outcomes". Can you please elaborate which design steps you have taken to achieve this prioritization?"*
>
> We would like to clarify that our tasks, specifically those relating to dialog, have reward functions that prioritize fairness and have been designed to be fairly intuitive. Tasks such as Twenty Questions and Guess My City have reward functions that are very neutral in nature such that you receive positive reward if you guess the name of the object or city. However for the Car Dealer task, the seller agent could potentially be deceptive or nefarious towards the buyer agent. In order to avoid unexpected behaviors, we have prompted both the seller and buyer agents with their respective strategies in order to generate the dataset (i.e. the buyer likes discounts), and have designed the reward function to be very straightforward (the price the car was bought for) and reflective of the outcome for both the buyer and the seller. We would like to clarify that our statement regarding dual-use implications of the work is an acknowledgement that there are unintended consequences from dialog generated with LLMs, including being persuasive, manipulative, etc. These risks would also be present in data that was collected from humans. We believe that finetuning LLMs with RL objectives can minimize such risks, and that research into RL/LLMs and such a benchmark can help us train LLMs that are more aligned with humans. We hope that our work at the intersection of RL and LLMs is one step in that direction. As per your comment, we will clarify our ethics section further with these points.
>
> 5. *You also express the intent to make public the code, datasets, hyperparameters as well as training procedure. What makes you confident that you sufficiently mitigate the stated implications and risks such that it is safe to publicly release the benchmark and open-source framework, instead of pursuing a different path such as making these more sensitive aspects of the results available only to trusted audiences (ie. known and trusted researchers, US AI Safety Institute, etc)?*
>
> We believe that by making the code, datasets, hyperparameters, etc. available, we provide a toolkit for researchers and practitioners to get started with multi-turn RL for LLMs (focusing on both online & offline RL). Specifically, we have trained models with fewer parameters specifically to allow for further development of algorithms by everyone regardless of access to large compute and resources, further emphasizing the importance of equity in opportunity to do research in RL and LLMs.

---

> > ### Author Response · Authors · 2024-12-04
> >
> > We hope these responses have helped answer the reviewers questions and that you consider raising your score :)

---

### Meta-Review · Area_Chair_3GsZ · 2024-12-19

**Metareview:**

This paper presents a new benchmark to evaluate LLM agents in a dialogue setting. An agent interacts with an LLM (a proxy for a human) to engage in a dialogue to solve an RL task.

Strengths:
This is an important and interesting task, and one that is surprisingly overlooked in LLM benchmarks. The more common tasks is for the LLM to generate a single response and get reward for it, or to take symbolic actions in multi-turn setting. However, reviewers raised several concerns regarding the benchmark:

Weakness:
1. Tasks are somewhat simplistic; not all are natural dialogue tasks. This was noted by the reviewer j29V and 9eFK. I personally found car dealer as an example of a good task and it would have been great to have more real-world tasks like it (e.g., hotel recommendation, flight booking).

2. Evaluations are restricted to GPT2 models. I understand that only a few labs can train GPT-4 or even 70B models, but GPT-2 is at this point quite outdated. Even a 2B or 3B model would have been nice.

Overall, I like this direction but I think this needs more work. At a minimum, either more real-world tasks or experiments with bigger models would be needed. Alternatively, authors can focus more on the human evaluation of LLM agents. For now, I am recommending a weak reject, but I wouldn't mind if the paper was accepted.

**Additional Comments On Reviewer Discussion:**

Reviewers focused on

1. Lack of experiments with models bigger than GPT2 (reviewer 9eFK and reviewer KbFz raised this)
2. Tasks being somewhat simplistic or not requiring language understanding (reviewer 9eFK and j29V)
3. Ethical concerns about whether the benchmark can be used improperly
4. Whether human study was properly done

(1) and (2) are indeed concerning. Regarding (3), I am unsure. I think most papers can benefit from ethics review so I am not against it but I think this paper is not using a realistic dataset and is not different from the vast majority of works in the LLM benchmark space. The emphasis on persuasion does put a bit different angle, so perhaps an ethics review can potentially help. Finally, the authors ran self-consistency experiments to validate the benchmark for (4) and the results look good to me so this issue was resolved.

---

### Decision · Program_Chairs · 2025-01-22

Reject